# Synthesis of a Magnetic Nanostructured Composite Sorbent Only from Waste Materials

**DOI:** 10.3390/ma16247696

**Published:** 2023-12-18

**Authors:** Alexander Bunge, Cristian Leoștean, Rodica Turcu

**Affiliations:** National Institute R&D for Isotopic and Molecular Technology, 67-103 Donat Street, 400293 Cluj-Napoca, Romania; cristian.leostean@itim-cj.ro

**Keywords:** sawdust, iron mud, magnetic nanomaterials, adsorption, dyes

## Abstract

Water pollution is a big problem for the environment, and thus depollution, especially by adsorption processes, has garnered a lot of interest in research over the last decades. Since sorbents would be used in large quantities, ideally, they should be cheaply prepared in scalable reactions from waste materials or renewable sources and be reusable. Herein, we describe a novel preparation of a range of magnetic sorbents only from waste materials (sawdust and iron mud) and their performance in the adsorption of several dyes (methylene blue, crystal violet, fast green FCF, and congo red). The preparation is performed in a hydrothermal process and is thus easily scalable and requires little sophisticated equipment. The magnetic nanostructured materials were analyzed using FTIR, VSM, SEM/EDX, XRD, and XPS. For crystal violet as a pollutant, more in-depth adsorption studies were performed. It was found that the best-performing magnetic sorbent had a maximum sorption capacity of 97.9 mg/g for crystal violet (methylene blue: 149.8 mg/g, fast green FCF: 52.2 mg/g, congo red: 10.5 mg/g), could be reused several times without drastic changes in sorption behavior, and was easily separable from the solution by simply applying a magnet. It is thus envisioned to be used for depollution in industrial/environmental applications, especially for cationic dyes.

## 1. Introduction

Especially in recent decades, the protection of the environment has become one of the most important issues that society as a whole faces. This broad term encompasses not only aspects of remediation of already polluted areas but also preventative aspects such as the cessation of using non-renewable sources as raw materials [1] and avoidance of waste wherever possible [2]. For the latter, the concept of circular economy has been formulated [3], which causes a paradigm shift towards the avoidance and/or reutilization of waste in production chains. In industrial application of chemical reactions, “Green Chemistry” is more and more used, which, among other things, mandates also the avoidance of waste, the use of renewable resources, as well as atom economy and the use of safe solvents [4].

For polluted water bodies, either streams or lakes in nature, or waste streams generated by man-made industrial processes, many different ways for remediation are currently used and being further researched, such as sedimentation, flocculation, [5] reverse osmosis, [6] or filtration [7]. However, the method of waste water treatment that is currently most researched is adsorption [8]. Adsorption requires a sorbent that is spent after its use in water treatment; however, the enormous diversity of potential materials to be used as sorbent ensures that this method can be used for and adapted to most of the different types of water contamination that exist [9]. Specifically, the use of waste material [10] and here particularly from renewable sources such as plant matter [11,12] is an important subject of research as, besides playing into the concept of circular economy, these materials are also available in large quantities and for a low price, which is necessary to make them competitive in this field. Furthermore, many sorbents can also be reused later on, [13] further decreasing the economic and ecological impact of producing a specific sorbent. Sawdust is one such biomaterial that has been used frequently, either directly as a sorbent or as a reactant to prepare new sorbents [14]. Sawdust is produced in large quantities and in many places as waste material, and as such is easy to obtain in large quantities without spending additional effort, and with low transport costs. It has thus been used as sorbent for pollutants: heavy metals such as cadmium [15] or chromium, [16] dyes such as direct brown [17] or methylene blue [18], as well as antibiotics, [19] among others.

Magnetic sorbents are another subtype of sorbents [20]. These consist either only of magnetic material, usually as nanoparticles so that they possess a large surface with which to adsorb pollutants [21], or as a composite with another nonmagnetic sorbent [22,23]. In the latter case, the advantage would be combining the superior sorption characteristics of the nonmagnetic sorbent with magnetism. Magnetism would allow the easy separation of the sorbent after the sorption process, ideally to be reused again after a desorption process. Magnetic composites have also been researched as sorbent for many pollutants, such as for heavy metals, [24,25] dyes, [26,27] or oil [28,29].

As stated above, one main advantage of adsorption as a process of water depollution is its adaptability. Many different categories of pollutants exist, such as heavy metals, [30] pharmaceutics, [31] or chemical waste products. Among these, dyes are a major source of polluted water [32]. The majority of dyes are used in the textile industry; however, significant amounts are also used in other industries, such as food production, the medical field, or cosmetics [33]. Depending on their target application, dyes can have different properties, [34] especially concerning hydrophilicity/hydrophobicity and whether they are anionic, cationic, or neutral. These properties greatly influence their sorption behavior regarding different sorbents. Crystal violet is a cationic triphenylmethane dye for which adsorption has been studied in detail using different classes of sorbents such as biomaterials [35] (including sawdust [36]), magnetic nanoparticles [37] or inorganic-organic composites [38].

While a large amount of research has been performed in preparing magnetic sorbents, both as nanoparticles and composites for different classes of pollutants, in most cases, the magnetic part was either separately or in situ prepared from metal salts [39]. Even when a waste product was used as the nonmagnetic part of the composite, this still only partially fulfills the principle of using only waste or renewable sources in the preparation of new materials. In some cases, waste was also used as a source for the magnetic part [40]. There exists a variety of iron-containing wastes that can serve as an excellent starting material for magnetic sorbents, such as red mud, [41,42] sewage sludge, [43,44] or electroplating sludge. Among these, the sludge obtained from aeration of ground water is especially suitable for this purpose, as it contains high amounts of iron, with small amounts of manganese as the main contamination. This iron mud obtained from groundwater treatment has been used in several publications for preparation of a magnetic sorbent (mostly in solvothermal or hydrothermal reactions), [45,46,47,48,49] however in each case also additional chemicals from nonrenewable sources (organic solvents or additional inorganic acids or bases) were used.

In this work, new magnetic composites have been synthesized from sawdust, [50] a waste material from renewable sources, and iron mud obtained from groundwater treatment, using only water as an additional component in the reaction. They, therefore, fully fulfill the requirements of circular economy and green chemistry to use only waste products and that any solvents used should be non-toxic. This is the first time a magnetic sorbent has been prepared only using waste materials and water. The materials have been characterized using scanning electron microscopy/energy dispersive x-ray spectroscopy (SEM/EDX), a vibrating sample magnetometer (VSM), Fourier-transform infrared spectroscopy (FTIR), x-ray powder diffraction (XRD) as well as x-ray photoelectron spectroscopy. Their adsorptive capabilities were tested using four dyes: methylene blue (MB), crystal violet (CV), fast green FCF (FG), and congo red (CR). These dyes represent multiple classes of dyes: anionic (FG, CR) and cationic dyes (MB, CV), azo (CR), triphenylmethane (FG, CV), and thiazine dyes (MB), used in medicine (MB, CV), food industry (FG) and textile industry (CR, CV). Further, MB is extensively used as a test pollutant in the synthesis of sorbents, and thus, the results can be compared with a large variety of literature data. Crystal violet has been used as a stain in medicine, as well as a textile dye. Due to its low biodegradability and toxicity, water sources contaminated with CV should be purified before being released into the environment. Thus, the suitability of the material with the best characteristics as a sorbent regarding CV has been researched in more detail by studying kinetics, isotherms, and behavior at different pH levels as well as different temperatures and desorption/reuse.

## 2. Materials and Methods

Oak sawdust **1** was a gift from Carmangeria Muresan, Cluj-Napoca, Romania. Iron mud from groundwater treatment **2** was a gift from Berliner Wasserbetriebe, Berlin, Germany. The iron mud was obtained by aeration of ground water, collected as a suspension in water, and air-dried prior to use. Crystal violet was obtained from Loba Chemie Fischamend, methylene blue from Merck, Germany, fast green FCF from Alfa Aesar (Thermo Fisher, Kandel, Germany), and congo red from Reactivul Bucuresti (Bucharest, Romania). All other chemicals were purchased from Sigma Aldrich (Taufkirchen, Germany). The neodymium magnets used were purchased from Euromagnet, Cluj-Napoca, Romania, and had dimensions of 5 × 5 × 2.5 cm, type N48.

### 2.1. Synthesis of Magnetic Composites 3

Saw dust **1** and iron mud **2** were mixed together with water (40 mL) and stirred lightly with a glass rod. The mixture was then put into a Teflon-lined autoclave and heated at a temperature T for a certain amount of time t. After cooling down naturally, the nanomaterials **3** were washed with water and separated by a magnet (left to separate on the magnet for ca. 15 min) before being dried over night in an oven (333.15 K) to give black solids. The detailed reaction conditions for the specific samples are listed in Table 1.

### 2.2. Preliminary Adsorption Tests

Stock solutions of 5 mg/mL for each of the dyes MB, CV, FG, and CR were prepared, and calibration curves were recorded using a UV/VIS spectrophotometer. For this, the absorbances at 665 nm (MB), 625 nm (FG), 590 nm (CV), and 500 nm (CR) were recorded, respectively (Appendix A). In the case of CV, a calibration curve in acetone was performed as well.

Magnetic separation was performed in two steps, first, the bulk of the sorbent was removed from the solution by a neodymium magnet within ca. 1 min, while any remaining sorbent (<1%) was removed over an additional 30 min.

For the adsorption tests, m_sorbent_ = 20 mg of the sorbents **1**, **2,** and **3** were stirred magnetically in an Erlenmeyer flask with V = 20 mL of a C_0_ = 0.25 mg/mL solution of the respective dye. After 17 h, the sorbent was separated magnetically, and the concentration of the dye C_e_ was determined in the supernatant. The sorption capacity q_e_ was then determined by
q_e_ = V ∗ (C_0_ − C_e_)/m_sorbent_(1)

### 2.3. Kinetics Experiments

The kinetics tests were performed as described before, except that the adsorption time was varied. The obtained experimental data points were then attempted to fit according to a pseudo-first-order and pseudo-second-order model as well as three different types of intraparticle diffusion models (for the formulae, see the Appendix A).

### 2.4. pH-Dependence of Adsorption

For the pH dependence, m_sorbent_ = 20 mg of **3d** was stirred in an Erlenmeyer flask with V = 20 mL of a C_0_ = 0.125 mg/mL solution of crystal violet in phosphate-citrate buffer (see Appendix A, both text and Appendix A). After 17 h, the sorbent was separated magnetically (see also Section 2.2), and the concentration of the dye C_e_ was determined in the supernatant.

### 2.5. Isotherm Determination Studies

The adsorption tests were performed as follows: m_sorbent_ = 20 mg of **3d** was stirred in an Erlenmeyer flask with V = 20 mL solution of crystal violet with varying concentrations C_0_ of CV. After 17 h, the sorbent was separated magnetically (see also Section 2.2), the concentration C_e_ was determined by UV-VIS measurement, and the removal efficiency was calculated as (C_0_ − C_e_)/C_0_. The sorption capacity q_e_ was plotted against C_e_, and the obtained data points were fit according to the Langmuir and Freundlich isotherm models (see Appendix A).

### 2.6. Thermodynamic Parameters

Adsorption tests were performed as follows: m_sorbent_ = 20 mg of **3d** was stirred in an Erlenmeyer flask with V = 20 mL of a C_0_ = 0.125 mg/mL solution of crystal violet at different temperatures. After 16 h, the sorbent was separated magnetically (see also Section 2.2), and the concentration C_e_ was determined by UV-VIS measurement. The obtained values were plotted as ln(K°) vs. 1/T (see also Appendix A; K°: equilibrium constant for the sorption process), and a linear fitting was performed to obtain values for ΔH° (enthalpy change) and ΔS° (entropy change).

### 2.7. Reusability Studies

The sorbent (100 mg) was placed in an Erlenmeyer flask and stirred with the CV solution (100 mL, 0.05 mg/mL) for 17 h. After magnetic separation, the sorption capacity q_e_ was determined as described above. For desorption, acetone (20 mL) was then added to the sorbent, and the mixture was stirred for 4 h. The supernatant was separated magnetically, and the sorbent was washed with water (20 mL) before being subjected to the next cycle of adsorption/desorption. The concentration C_e_′ of CV in acetone was then determined from the calibration curve, and the desorption capacity q_e_′ was determined according to the formula
q_e_′ = V ∗ C_e_′/m_sorbent_(2)

### 2.8. Characterization

The morphology of the samples was characterized by Scanning Electron Microscopy (SEM) using a Hitachi SU8230 Scanning Electron Microscope, Tokyo, Japan, and the elemental analysis EDX used an Energy-Dispersive Spectroscopy (EDS) detector X-Max 1160 EDX (Oxford Instruments, Oxford, UK). Samples were mounted on the specimen holder using double adhesive carbon discs to ensure proper electrical conduction and were examined in high-vacuum mode at an acceleration voltage of 30 kV. Magnetic hysteresis measurements were recorded using a Cryogenic VSM magnetometer at room temperature with a maximum field of 5T. Fourier transform infrared (FTIR) spectra were recorded using a JASCO FTIR 4600A spectrophotometer with ATR-PRO-ONE accessory, CO_2_-, H_2_O-, ATR- and baseline-corrected as well as smoothed and normalized for better visibility of the bands.

UV-VIS spectra were taken using a Jasco V-550 UV-VIS Spectrophotometer (JASCO Deutschland GmbH), equipped with a double-beam photometer and a single monochromator, using 10 mm length quartz cells. X-ray power diffraction (XRD) measurements were performed with a Bruker D8 Advance X-ray diffractometer, with a Ge (111) monochromator for Cu-Kα1 radiation (λ = 1.5406 Å) having the source power of 40 kV and 40 mA, at room temperature and LynxEye position-sensitive detector. A SPECS XPS spectrometer equipped with an Al/Mg dual-anode X-ray source, a PHOIBOS 150 2D CCD hemispherical energy analyzer, and a multichanneltron detector with vacuum maintained at 1 × 10^−9^ Torr was used to record the XPS spectra. The Al K_α_ X-ray source (1486.6 eV) was operated at 200 W. The XPS survey spectra were recorded at 30 eV pass energy and 0.5 eV/step. The high-resolution spectra for the individual elements were recorded by accumulating 10 scans at 30 eV pass energy and 0.1 eV/step. Data analysis and curve fitting were performed using CasaXPS software with a Gaussian-Lorentzian product function and a nonlinear Shirley background subtraction. Peak shifts due to any apparent charging were normalized with the C1s peak set to 284.8 eV. The high-resolution spectra were partly deconvoluted into the components in order to determine the particular bond types present at the sample surface.

## 3. Results and Discussion

### 3.1. Synthesis of the New Materials

The magnetic nanostructured materials **3** were synthesized from saw dust **1** and iron mud **2** by a hydrothermal reaction in water under different conditions (Figure 1). Iron mud, by its nature of being obtained by air oxidation of ground water at room temperature, contains iron only in a trivalent state and in an amorphous form [51]. In order to prepare magnetic materials from it, a reductant needs to be added to the reaction mixture [47,48]. In the case of materials **3**, this reductant is obtained from saw dust. Sawdust, as a wood waste product, consists mainly of the natural polymers cellulose, hemicellulose, and lignin. During the solvothermal treatment, depolymerization and solubilization processes take place, which produce mono/oligosaccharides that can serve as reductants due to their free aldehyde groups [52,53]. Sawdust is a cheap and ubiquitous waste material and thus has been researched as sorbent in many cases [14,17,54,55]. In this reaction, it thus serves a double role as starting material for the hydrochar that makes one part of the sorbent material and as a reductant for the iron mud to make the final material magnetic without having to add another reagent.

The amounts of sorbent obtained in each case (Table 2) are lower than the sum of the amounts of starting materials added to the reaction mixture. The main reason for this is the aforementioned solubilization of parts of the sawdust. It can also be seen (comparing **3a**–**c**) that the amount of sorbent obtained is only weakly dependent on reaction time, but (comparing **3a** and **3d,f**) is highly dependent on the reaction temperature, decreasing with lower temperature. Because the magnetization of the sorbents also decreases with diminished temperature, it is likely that this loss in yield is due to the magnetic separation process, where nonmagnetic material would get washed away. An imbalance in reactants (**3g**,**h**, i.e., either a lower or higher mass of either sawdust or iron mud compared to the other reactant) also causes some loss of product mass, likely for the same reason, while interestingly, both a higher (**3j**) and lower (**3i**) concentration yield a somewhat increased relative amount of product.

### 3.2. Magnetization

The magnetization curves were measured for all samples **3** at room temperature (Figure 1, Table 2). Compared to bulk magnetite (M_s_ = 92 emu/g [56]), the saturation magnetization is relatively low; however, it is still appropriate for its intended application and comparable to or better than other works [45,47,48] using a wet chemical approach to produce magnetic materials from iron mud without completely dissolving it in acid first [57]. The saturation magnetization is highest for a reaction temperature of 473.15 K (**3a**,**b**); with lower temperatures, M_s_ (**3d**–**f**) strongly decreases. A longer reaction time (**3b**) had little influence on M_s_; however, there was a noticeable decrease when using shorter reaction times (**3c**). For an imbalanced amount of iron mud and sawdust (**3g**,**h**), M_s_ decreased notably/slightly, while for lower (**3i**) or higher concentration (**3j**), the saturation magnetization increased/decreased slightly compared to **3d**. During the reaction there are likely two important steps happening that affect the saturation magnetization: solubilization of reducing compounds from sawdust and reduction in ferric ions to form magnetite. The decrease in M_s_ when a shorter reaction time was employed is likely because there was not enough time for one or both of the important steps to be completed. Since the total mass of product for **3c** is almost the same as for **3b**, and a mass loss could be expected due to the magnetic separation (see Section 3.1, last paragraph), it is likely that already the solubilization step is not completed. Both steps are basically complete after 17 h, so a longer reaction time will not influence the magnetization much. Lowering the reaction temperature will cause a similar effect as decreasing the reaction time, although in this case, it is more likely that the reduction in ferric ions is the step that cannot take place to the full extent. This can be concluded from the fact that the amounts of the product also decreased significantly (Table 2), more than for the shortening of reaction time (**3c**), and also because it is known that the temperatures needed for reduction in ferric to ferrous ions for polyol [58] and hydrothermal reactions [59,60] are higher (depending on the reaction conditions about 433.15–473.15 K). In the case of imbalanced quantities of starting materials, an under- (**3g**) or overreduction (**3h**) can take place, decreasing the overall magnetization of the material.

### 3.3. Morphology and Composition Analysis by SEM/EDX

All samples, including the starting materials **1** and **2**, were analyzed by SEM (Figure 2 and Appendix A). Sawdust **1** had the typical structure of wood, [61] a fibrous structure which was highly porous. In contrast, unmodified iron mud consisted of polydisperse and nonuniform particles up to micrometer size, without visible pores. After the reaction, the newly obtained materials still kept the overall shape of wood fibers, upon which were deposited the iron-containing particles. Neither the wood fibers nor the particles changed their shape for either of the reaction conditions tested. However, since the deposition of particles on the wood fibers was ubiquitous, it is clear that the sorbents **3** are actually distinct new materials and not just a mixture of two components that could be split back into its components again. The partial dissolution of hemicellulose, which happens during the reaction, is not visible in the morphology of the materials, and the shape retention of the iron-containing particles means that the reaction mechanism cannot go via dissolution-reprecipitation. This also makes it likely that any phase transformations (including reduction in ferric to ferrous ions) in the iron-containing phases will only happen on the surface of the particles and not in the bulk, which can explain the relatively low saturation magnetization compared to bulk magnetite. All new materials **3** looked similar; the main differences were in **3g** and **3h**, which had a comparatively higher/lower ratio of fibrous structure/particulate structures.

EDX was performed for starting materials **1** and **2**, as well as for samples **3d**,**h** (Figure 2 and Appendix A; only one spectrum per sample is shown; the other spectra look similar). Sawdust **1** is composed mainly of carbon and oxygen. With the exception of a very small amount of calcium, no other elements were detectable in concentrations above 0.1%. From the EDX spectrum of iron mud **2**, it is visible that besides iron, only small amounts of Mn and Ca are detectable in the iron mud sample. The manganese precipitates in the same way as iron by oxidation of Mn(II) ions in ground water during the aeration. Calcium likely precipitates as carbonate because of CO_2_ from the air used for aeration. It is interesting that unlike in other sources that researched iron mud, no other elements such as Si or Al could be detected. This means that the iron mud, despite being a waste product, has a high iron content and is thus very suitable as a source for the production of magnetic materials. The ratios of Mn/Fe and Ca/Fe determined by EDX are displayed in Appendix A. These ratios are comparable between samples **3d**,**h**, but lower than for the raw material **2**, especially in regard to the ratio Ca/Fe. This means that during the reaction to prepare materials **3**, an additional purification process takes place, which eliminates most of the nonmagnetic Ca and some of the Mn, increasing the potential saturation magnetization of the new composites **3**.

### 3.4. FTIR-Studies

In order to evaluate functional groups, FTIR was performed for starting materials **1** and **2** as well as composites **3** (Figure 3 and Appendix A).

The bands of sawdust **1** were typical for wood samples, [62] with a large band at 3000–3500 cm^−1^ corresponding to ν(O-H), small bands at 2845 and 2922 cm^−1^ indicative of ν(C-H), bands at 1669 and 1621 cm^−1^ matching ν(C=O) of carboxyl groups and a band at 1021 cm^−1^ from ν(C-O) groups the most prominent ones. In this regard, it is interesting to note that the O-H band is of relatively low intensity compared to the C=O bands, unlike in other research works [63]. This could mean that sawdust **1** already contains a large number of carboxyl groups, which can help in the adsorption of cationic species. Iron mud **2** also has an FTIR spectrum similar to those published before, [64] with a large band at 3000–3500 cm^−1^ corresponding to ν(O-H), bands at 1650–1380 cm^−1^, and possibly 953 cm^−1^ likely due to adsorbed carbonate, and a band at 418 cm^−1^ from the stretching vibrations of Fe/Mn-OH.

The bands for samples **3** are similar to each other, most corresponding to the bands from sawdust **1**. There is, however, a new band at 553 cm^−1^ in almost all samples, which is typical for ν(Fe-O) in magnetite. The band at 418 cm^−1^ is still visible, however, and the band at 3000–3500 cm^−1^ is of higher intensity than in sawdust **1**. This indicates that iron mud was not completely transformed into magnetite. The spectrum of **3f** shows some differences: Thus, the bands between 1700–1300 cm^−1^ are broader, and the band around 1000 cm^−1^ has a shoulder to lower wavenumbers. This is similar to the corresponding bands in iron mud and shows that at this low temperature, carbonate desorption has not yet taken place, and thus, these bands overlap with the ones from sawdust. The magnetite band at 553 cm^−1^ is also barely visible, which is (besides the low magnetization of the sample) another indicator that magnetite has only formed in very small quantities at this temperature.

### 3.5. Preliminary Sorption Tests

The prepared samples **3,** together with the starting materials sawdust **1** and iron mud **2** were then subjected to sorption tests with four different dyes as pollutants. The focus of this work is on the adsorption of crystal violet; however, the other dyes were chosen for multiple reasons: Methylene blue is a very common model pollutant to test new sorbents, thus this work can be compared to a larger range of other published works. Additionally, the four different dyes represent different structural classes (thiazine dye, methylene blue(MB); triphenylmethane dye, crystal violet(CV) and fast green FCF(FG); azo dye, congo red(CR)), industrial uses (medical field, MB and CV; textiles, CR and CV; food industry, FG) and physicochemical properties (cationic dye, MB and CV; anionic dye, FG and CR). With these results, it is possible to make at least tentative predictions for the sorption behavior of other pollutants.

The results of initial sorption capacity determination for the four dyes for all samples **3,** as well as the starting materials **1** and **2,** are displayed in Figure 4. The formulae for MB, CV, FG, and CR are shown in Figure 2. It is apparent that for all dyes, with the exception of CR, at least some of the materials **3** have a higher sorption capacity than either of the starting materials saw dust **1** and iron mud **2**. For CR, iron mud **2** shows by far the highest sorption capacity. There are several general mechanisms with which dyes could adsorb to sorbents: the electrostatic attraction between the positively charged dye molecule and negatively charged sorbent or vice versa, H-bonds, van der Waals or π-bonding, hydrophobic interactions or steric reasons. It is not possible to explain the adsorption behavior in the case of CR only with electrostatic interactions (CR is negatively charged due to its sulfonate groups, and both materials **1** and **2** have been proven by FTIR to also have negatively charged carboxylate/carbonate groups), as for the other anionic dye, FG, **2** exhibits one of the lowest sorption capacities. Hydrogen bonds, van der Waals or π bonding, as well as hydrophobic interactions, generally offer a lower attraction force than electrostatic interactions, so these would likely play only a supporting role, if at all. Thus, the explanation for this behavior must lie in the structural differences, either due to the functional groups (azo vs. triphenylmethane) or in the geometry of the structures. For the three dyes MB, CV, and FG, magnetic composite **3d** shows the highest sorption capacity, which is why we chose to pursue further studies with this material.

### 3.6. XRD Studies

Sample **3d** was then analyzed by XRD and compared to precursor **2** (Figure 5). Iron mud **2** is basically amorphous; it is maybe possible to assign a small peak to calcite; however, this is no definite proof. In contrast, sample **3d** shows broad reflections for 2-line ferrihydrite, together with small peaks that can be assigned to magnetite. Just from XRD, the differentiation between magnetite and other spinel structures (especially maghemite) cannot be made; however, together with the results from FTIR and XPS (see below), it can be concluded that magnetite is indeed present in the sample. No peaks relating to calcite can be found. These results show that while the morphology of the iron-containing particles does not change, their structure does change towards both 2-line ferrihydrite and some magnetite. The presence of a still large amount of ferrihydrite can explain the relatively low saturation magnetization compared to bulk magnetite (92 emu/g). The absence of any signs of calcite corroborates the findings from EDX that some calcium was removed during the synthesis of the magnetic nanocomposites.

### 3.7. XPS Analysis of Composite 3d

The magnetic nanocomposite **3d** was subsequently also analyzed by XPS for further clarification of composition and structure (Figure 6 and Appendix A). In the survey spectrum, only the expected elements of carbon, oxygen, iron, manganese, and calcium are visible. The ratios of Mn/Fe and Ca/Fe are shown in Appendix A. These ratios differ to some extent from the ones determined by EDX. The likely reason for this discrepancy is that XPS is a surface analysis method. This means that both calcium and manganese are more likely to occur on the surface of the synthesized sorbent. The Fe2p3/2 spectrum can be deconvoluted into several species and shows the presence of both tetrahedral and octahedral ferric as well as octahedral ferrous ions at 714.3, 712.5, and 710.6 eV, respectively. The ratio between ferrous and ferric ions is 0.95, which is different from the ideal ratio of 0.5 for magnetite and actually higher than could be expected from a mixture of ferrihydrite and magnetite that XRD suggested. The reason for this is, again, that XPS is a surface method. From the results, one can see that the iron mud particles have only been reduced on the surface, while the core retained ferric ions in the form of ferrihydrite. This also confirms the observation from a comparison of the morphologies that the reduction did not occur via dissolution of the iron-containing phase but rather a transformation as well as a surface reduction. In reduction reactions similar to a polyol process, it is not uncommon that the amount of ferrous ions is somewhat higher than necessary for a stoichiometric formation of magnetite [65]. These additional ferrous ions on the surface likely form amorphous structures and thus cannot be identified by XRD.

The C1s spectrum shows the expected peaks of C-C (mostly from lignin) and C-O at 284.9 and 286.6 eV, respectively. Additionally, a peak at 288.5 eV that can be attributed to either carbonyl- or carboxyl groups proves that significant oxidation of the sawdust took place during the reaction, confirming the results obtained by FTIR. The carboxylic groups likely help adsorb cationic dyes (MB and CV), as demonstrated in the initial sorption tests (Figure 5). In the O1s spectrum, contributions from C=O and C-O at 533.2 and 531.3 eV confirm the observations of the C1s spectrum, while a peak at 530.0 eV stems from metal-oxygen (mostly Fe-O) binding (Appendix A). Additionally, a Ca2p spectrum was taken, with peaks at 351.1 and 347.5 eV corresponding to Ca2p1/2 and Ca2p3/2, respectively (Appendix A). The binding energy of these peaks corresponds well with that of calcium carbonate; [66] however, there is an additional peak at lower binding energy (344.8 eV). In other literature sources, [66,67,68,69] a peak at such low binding energy was often attributed to either metallic calcium or some mixed metal oxides. Since, under the reaction conditions, metallic calcium is unlikely to form, probably a mixed oxide, such as calcium ferrite or maganite, is the source of this peak.

### 3.8. Effect of Contact Time

After the structural and morphological properties of the sorbents had been determined, additional effort was made to determine the mechanism and properties of the sorption process. For this, kinetic studies were first undertaken (Figure 7).

The data points for the sorption capacity obtained by varying the contact time were attempted to be fitted with curves corresponding to pseudo-first-order and pseudo-second-order kinetics, as well as the intraparticle diffusion model (see also Appendix A). In order to have a more accurate fitting, only the nonlinear forms of the curves were used [70], and orthogonal distance regression was used for the iteration steps to account for the fact that there can be errors both in the determined values of q_t_ as well as t. The corresponding values for R² are 0.9999 for the pseudo-first-order, 0.99995 for the pseudo-second-order, and 0.998/0.9998 for the intraparticle diffusion model with one/multiple adsorption processes. A plot of q_t_ vs. t^0.5^ was also done, and the experimental data were separated into three sets of linear functions (Appendix A). The R² values for these functions were 0.974, 0.996, and 0.997, respectively.

In the literature, many sorption processes are said to proceed according to a pseudo-first-order kinetic, or Lagergren model [71]. However, there have been reports that many sorption processes described as pseudo-first-order kinetic are actually pseudo-second-order processes [72]. On the other hand, it has also been argued that a pseudo-first-order model describes many data sets better if a different method for fitting is used [70]. Both models assume that the sorption process is the rate-limiting step; while for pseudo-first-order kinetics, the sorbent is seen as having no specific adsorption sites, for pseudo-second-order kinetics, a distinct and finite number of sorption sites is assumed. In contrast, the intraparticle diffusion or Weber-Morris model has the premise that the rate-limiting step, as its name suggests, is a diffusion process. For this model, multiple variants exist, splitting the sorption process into one, two, or three parts.

Among the pseudo-first-order, pseudo-second-order, and the first two intraparticle diffusion models, the fitted curve for pseudo-second-order kinetics has the best R² value and also looks graphically closest to the obtained data points. The intraparticle diffusion model assuming three different processes could not be fit using a plot of q_t_ vs. t. For this reason, and for having to use a different iteration process, the obtained R² values cannot be compared directly to the other obtained values, and thus, a conclusion about whether or not it is a better fit than the pseudo-second-order model cannot be reached by this. In our interpretation, it cannot be excluded that intraparticle diffusion in this manner is the rate-limiting step; however, because for this model, the available set of data points has to be split in three, increasing the uncertainty of the fitting, and because neither of the other two intraparticle diffusion models describes the experimental data even close to as good as the pseudo-second-order kinetics model graphically, the latter model was preferred. Thus, for magnetic sorbent **3d** and CV, the pseudo-second-order model describes the sorption process better. The corresponding parameters are q_e_ = 71 mg/g and k_2_ = 0.0093 min^−1^. Based on the assumption that the time dependency follows a pseudo-second-order model best, the mechanism of adsorption would then be a chemisorption process. While the word chemisorption has generally been used in the literature when describing the implications following kinetics adhering to a pseudo-second-order law, in this case, the term does not necessarily mean adsorption via the formation of covalent bonds but merely stronger bonds than would be formed through physisorption processes. This means that the sorbent possesses specific adsorption sites to which the sorbate binds.

Other kinetic models were not taken into account. The main possibilities for rate-determining steps would be external mass transfer, diffusion, and adsorption processes. Agitation during the sorption process prevents the external mass transfer from playing a role as a rate-determining step, so it is to no avail including the external mass transfer model. Any other kinetic models based on diffusion and adsorption processes are more complicated than the ones included here and thus should only be tested if no good fit is reached with the models used in this work already.

### 3.9. pH Dependency of the Sorption Capacity

The sorption capacity of CV on **3d** was then followed over a range of pH from 3 to 8. This range was chosen for several reasons. Firstly, the buffer solution used (citrate-phosphate buffer) has its best capacity in this range, and a change in buffer might influence the sorption behavior. Different ions present in different types of buffer could adsorb to the sorbent with variable strength, thus potentially blocking sorption sites to the pollutant in an undesirable and unpredictable manner. Second, the investigated range is one where any real samples would most likely fall, as natural water bodies seldom have a too-low or too-high pH, and any industrial effluents would have to be neutralized before being released into the environment. Third, CV is known to desorb at low pH [73,74], and thus, a medium of low pH would be unsuitable to use most sorbents in; aside from that, at a low pH, the iron-containing part of the sorbent would dissolve. A sorption behavior like the one that CV is known for generally occurs in sorption processes with sorbents that exhibit anionic groups and pollutants with cationic groups or vice versa. For a low pH, the anionic groups would get protonated, limiting the amount of electrostatic attraction possible between sorbent and sorbate.

The sorption behavior of CV on **3d** at different levels of pH is shown in Figure 8. As can be seen, for most of the range of pH (4–8) the sorption capacity q only changes insignificantly, between 88.4 and 93.3 mg/g. Only at pH 3 is a decrease to 79.0 mg/g (by 11%) notable. This is most likely due to the aforementioned range where CV usually desorbs. The reasons for that are likely electrostatic in nature. Magnetic sorbent **3d** exhibits carboxylic acid groups, as seen from FTIR and XPS. At a more neutral pH, these groups are at least partly deprotonated, leading to negatively charged sorbent. This will bind CV (which is positively charged) electrostatically to the sorbent and thus increase the sorption capacity.

In general, it can be said that between pH 4–8, CV is adsorbed equally well on the magnetic sorbent **3d**.

### 3.10. Effect of Initial CV Concentration and Isotherms

The effect of different initial concentrations of CV on adsorption was determined for **3d,** and the removal efficiencies as well as sorption capacities were determined (Figure 9a and Appendix A). It can be seen that lower concentrations of pollutant (CV) result in higher removal efficiencies (up to 93% for C_0_ = 20 mg/L), but at higher concentrations of 150 mg/L, the efficiency is still quite good (56%). The decrease in removal efficiency at higher initial concentrations is to be expected [38] and supports the hypothesis that the material has a finite number of adsorption sites (see Section 3.8). The maximum removal efficiency of 93% is very good and demonstrates that **3d** can make an efficient sorbent for CV. For the sorption capacities, the trend is the opposite; at higher initial concentrations, q_e_ increases. The growth then slows down for higher initial concentrations of pollutants, which is also commonly seen in sorption curves [75].

To further determine the mechanism of adsorption, a plot of q_e_ vs. C_e_ was done, and the functions for Langmuir and Freundlich isotherms were attempted to be fit (Figure 9b). For the same reason as for the kinetic model fits, also in this case, only the nonlinear functions were used, and the iteration mechanism was orthogonal distance regression as well. The R² values were 0.994 for the Langmuir function and 0.991 for the Freundlich fit. Based on these values, as well as a visual evaluation by eye, it was decided that the Langmuir function fits the experimental data best. The obtained parameters from the fitting were q_0_ = 87.0 mg/g and K_L_ = 0.108 L/mg. The Langmuir adsorption model assumes that all sorption sites are equivalent and there will be only one layer of pollutant adsorbed on the sorbent. These assumptions fit well with the conclusions from kinetics studies. From FTIR and XPS, it was found that the sorbent contains carboxylate groups. CV is a cationic dye, which means that electrostatic interaction is a possible mechanism of sorption. In this case, only one molecule of CV would interact with one carboxylate group on the sorbent (because after the adsorption of this one molecule, the negative charge would be balanced). This sorption mechanism would fit (or at least not contradict) the assumptions made by the Langmuir model: the surface is homogenous in the sense that the sorption sites are only carboxylate groups, CV would then be immobile (also correlating with the fact that the intraparticle diffusion model does not fit well with the data obtained from kinetics experiments), all sorption sites are energetically equivalent, only one molecule of CV can be adsorbed per carboxylate group, and no or equal interactions exist for all adsorbed CV molecules.

### 3.11. Effect of Temperature

In order to determine the thermodynamic parameters of the adsorption of CV on **3d**, the sorption capacity was measured at different temperatures (Figure 10a). It is interesting to note that at lower temperatures, q_e_ increases quite drastically to 123 mg/g. The calculations are described in the Appendix A. Ln(K°) was plotted against 1/T, and a linear fit was performed that showed an R² of 0.997 (Figure 10b). The respective standard enthalpy and entropy changes determined are ΔH° = 36.8 kJ/mol and ΔS° = −39.7 J/(mol*K); the values of standard free enthalpy change at the different temperatures are displayed in Table 3.

The positive standard enthalpy change confirms that the reaction is endothermic. Because of the negative standard free enthalpy change, the sorption is spontaneous for all determined temperatures. The value of ΔH° of >20 kJ/mol makes it likely that the adsorption process is based on electrostatic interactions, as opposed to Van der Waals or similar interactions of low strength on the one hand, and covalent bond formation on the other hand [76]. This classification correlates with the structural information gained from FTIR/XPS as well as knowledge of the structure of the pollutant (CV), that the sorbent possesses negatively charged carboxyl groups, whereas CV contains positively charged amino groups. It also correlates with results obtained from the pH dependent studies (Section 3.9). Electrostatic interactions can be classified as physisorption, and this could, in theory, contradict results obtained from kinetics experiments, which suggest chemisorption (isotherm experiments suggest either chemi- or physisorption). However, the assessment of chemisorption from a pseudo-second-order kinetics model is only based on the fact that it means that a limited amount of adsorption sites exists (similar to the Langmuir model for isotherm studies). From this point of view, it also makes sense to define electrostatic interactions as chemisorption (in the sense in which it is used for pseudo-second-order kinetics). Other publications also shared this point of view [77,78]. Thus, ultimately, there is no contradiction between the results obtained by temperature and kinetics studies. In the literature, both endo- as well as exothermic processes are described for the sorption of dyes on biosorbents [79].

### 3.12. Desorption/Reusability

Since the quantities of a sorbent for a potential real-life application need to be large, a way to lower both the cost of production as well as the ecological impact that the produced waste would have is to reuse the sorbent. For this, it needs to be demonstrated that the pollutant (CV in this case) can be efficiently desorbed and that the reuse process is repeated a number of times. In the literature, there are several conditions for desorption given for a cationic dye like CV. Desorption has been performed using acidic conditions, [75] various organic solvents, [80] or brine [81].

After some preliminary tests, acetone was chosen for our experiment. The reasons for this are as follows: Acidic solutions have the possibility to react with the magnetic inorganic part (dissolution) and thus could possibly destroy the sorbent. Any aqueous solution, acidic or brine, would have to be disposed of again or regenerated, which, due to the relatively high boiling point of water, would require a larger amount of energy, if it was possible at all. Organic solvents, on the other hand, need to be reasonably cheap to produce, ideally also produced from renewable resources and biodegradable. Besides ethanol, acetone can also be produced from renewable resources, [82] biodegradable [83] and cheap. We ultimately chose acetone for desorption because it is able to efficiently desorb CV and also has a lower boiling point than ethanol. This way, the acetone solutions of CV can easily be regenerated by distillation at low temperatures. The results of several adsorption/desorption processes are displayed in Figure 11.

As can be seen, the sorbent **3d** desorbs, on average, about 80% of the adsorbed CV each cycle. The adsorption capacity gets slightly lower initially but then stabilizes at around 80% of the initial adsorption capacity, the removal efficiency of CV being about 86–67%. The notably lower desorption in cycle two is likely an experimental problem, as in cycle three, the desorption capacity is higher by an equivalent amount. The adsorption/desorption experiments were carried out with a relatively high amount of sorbent and a low concentration of pollutants. The reason for this is that in the targeted application (water depollution), the CV concentration would likely be comparatively low as well, and at the same time, as much CV as possible should be removed before the water (either an industrial effluent or a polluted waterbody) can be considered clean. The results show that the magnetic sorbent **3d** can be used repeatedly for CV desorption, with good removal efficiency.

### 3.13. Comparison with Other Sorbents

The sorption capacity of sample **3d** towards all four pollutants (MB, FG, CR, CV) was compared with data obtained from the literature (Table 4). There are many publications describing the adsorption of these four dyes, especially for MB; it is a type of reference sorbent for many authors. Because of this, the literature sorbents chosen for comparison were mainly magnetic, based on sawdust, based on lignocellulosic material, or a combination of these. It can be seen that in the case of MB and CV, nanocomposite **3d** compares favorably with many of the results obtained by the literature. For FG, **3d** was even better performing than all references displayed in this table. As discussed above, the sorption capacity of **3d** for CR is much lower than samples found in the literature; however, pure iron mud **2** performed almost as well as or better than the displayed literature references. For sorbent **3d,** the fact that it is magnetic and has been synthesized only from waste materials makes it a good choice as a sorbent for CV, MB, or FG depollution.

## 4. Conclusions

Magnetic nanomaterials have been synthesized from only sawdust, iron mud, and water using different reaction conditions. The different materials have been characterized using FTIR, VSM, SEM, and EDX, and their capacity towards adsorption of methylene blue, crystal violet, fast green FCF, and congo red has been tested. Optimum reaction conditions for obtaining a sorbent with the best sorption capacity, as well as other parameters like magnetization and amount obtained, have been determined, and the sorbent **3d** has been further characterized using XRD and XPS. It was found that neither the organic nor inorganic parts change much from a morphological point of view; however, more functionalities, especially carboxylic acid groups, are introduced to the organic part of the composite, which help in the adsorption of cationic dyes like MB and CV. The amount of introduced functional groups depends mainly on reaction temperature as well as time (until a certain point). For the inorganic part of the composite, a partial solubilization of noniron parts (calcium and magnesium) occurs, and ferric ions are reduced to ferrous ions on the surface, forming magnetite without a dissolution of the iron mud particles, while the core phase transforms into 2-line ferrihydrite.

The mechanism of adsorption of CV on sorbent **3d** was then investigated in more detail using time, concentration, pH, and temperature-dependent as well as reusability studies. It was found that the sorption process most closely resembles pseudo-second-order kinetics, which means the reaction likely proceeds sorption-controlled instead of diffusion-controlled, with chemisorption and having a finite number of adsorption sites. This last conclusion can be supported by the fact that the adsorption is most likely occurring according to the Langmuir model, which also states that a finite number of sorption sites exist, all of which can be occupied only once. From temperature studies, it was found that the sorption process happens spontaneously and is endothermic, with electrostatic interactions likely being involved in the sorption process. Since the sorption capacity is higher at lower temperatures, in a practical setting, it could be considered whether it would be feasible to cool down the polluted water body or effluent (by storing it, heat exchange, or waiting for more ideal outside temperatures) before subjecting it to the adsorption process. Moreover, the sorbent can be reused several times without significantly decreasing sorption capacity.

The sorbent compares well with other types of sorbent described in the literature and, thanks to its easy and scalable synthesis, would be a good candidate for a magnetic sorbent to remove pollutants for industrial/environmental applications.

## Data Availability

Raw data can be made available upon request to the authors.

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
