# Peer review of "Synthesis of a Magnetic Nanostructured Composite Sorbent Only from Waste Materials"

_materials, 2023, doi:10.3390/ma16247696_

Round 1

Reviewer 1 Report

Comments and Suggestions for Authors

This manuscript intends to provide a synthesis solution for a magnetic nanostructured composite sorbent made only from waste materials available in large quantities and at a low price. This manuscript is well organized. It could be accepted after a minor revision.

Line 68. Check the integrity of the sentence concerning the literature citation 24.

Line 136, do check C0.

Authors mentioned, "Sawdust is a cheap and ubiquitous waste material and thus has been researched as sorbent in many cases".The authors cited only one paper. Please provide more information and citations.

What magnet the authors used for this work should be noted because it may matter the yield. Do describe more details on the reclamation.

The authors mentioned the magnetite with Ms 92emu/g later on. Is it the magnet?

Line 216, "An imbalance in reactants", please define and clarify the imbalance. Besides this, the authors also mentioned "imbalance" several times; please detail them.

Line 243, check the sentence with the complete idea the author wants to express.

How to define the "distinct new" (line 258-260).

Please split this sentence into two to make it clearer.

Figure 3 and Figure S2 should also encompass results and discussion concerning all the spectra for illustration, not only one of them.

Lines 273-274, split this long sentence.

FTIR images are vague; please insert HR ones.

Line 294, delete the space before the "small".

Line 307, correct the writing mistakes.

SI line 33, unify the written form of all subscript of numbers.

Comments on the Quality of English Language

Fine presentation with the language English

Reviewer 2 Report

Comments and Suggestions for Authors

There are a lot of papers that have been published in this topic. The authors should emphasize how their current research is advancing the fundamental knowledge in the field.  A few relevant references should be added in the Introduction section of the manuscript:

·        https://doi.org/10.1016/j.jhazmat.2010.10.022

·        https://doi.org/10.1002/slct.202003606

·        https://doi.org/10.1016/j.biortech.2015.04.048

The reason for the XRD measurements is not apparent to this reviewer.

The figure captions in many cases would benefit from more descriptive texts.

Almost all of the adsorption data in figures should be repeated and presented with error bars.

The thermodynamic study must include at least one more temperature. Three data points is insufficient to draw conclusions (Figure 11).

If the mechanism of removal is chemisorption then why there is a need of applying intra-particle diffusion?

There is a slight contradiction in the manuscript where the kinetic results suggest adsorption follows the pseudo second order kinetics but the isotherm study show adherence to Langmuir isotherm.

The authors claim that the mechanism of chromium removal is chemisorption (Page 19) while isotherm reflects physisorption mechanism.  Please address this issue.

Why did the authors not look at other adsorption isotherms to explain the experimental results, such as the ones highlighted in review papers that focus on modeling of adsorption isotherm systems.

Comments on the Quality of English Language

Minor spelling and grammatical issues must be addressed.

Reviewer 3 Report

Comments and Suggestions for Authors

- Please add quantitative results in the abstract

- The introduction is short, please add the research gap of this study and some results obtained in the past, and the novelty of your work..

- Subsection from 2.4 to 2.8 please add more details information

- The correlation between reaction conditions and the magnetization of the materials is well-addressed. However, a clearer discussion on how changes in reaction temperature and time affect morphology and magnetization would add depth to the findings.

- The sorption capacity evaluation against different dyes and pH levels is intriguing. However, provide a clearer explanation of the mechanisms underlying the sorption process, especially regarding the interactions between the sorbent and different pollutants under varying pH conditions.

- The pseudo-second-order kinetic model fits the data well. However, discussing the significance of the chosen models and comparing their appropriateness for this study's context could strengthen the interpretation of results. Also, elaborate on the thermodynamic parameters obtained from the temperature-dependent sorption capacity.

- The Langmuir adsorption model fitting the experimental data well is an important finding. However, provide a deeper discussion regarding the assumptions made by the Langmuir model and how well it aligns with the experimental observations.

Comments on the Quality of English Language

Moderate editing of English language required

Reviewer 4 Report

Comments and Suggestions for Authors

This research work focused on synthesizing and studying the efficiency of Magnetic nanomaterials as ecological adsorbents. The studied adsorbent was synthesized from sawdust, iron mud, and water using different reaction conditions. According to the results of this study, the optimum reaction conditions of sorbent showed a significant adsorption sorption capacity. Through the study, the different materials have been characterized using FTIR, VSM, SEM, and EDX and their capacity towards adsorption of methylene blue, Crystal violet, fast green FCF, and congo red have been tested. The results of this study are interesting. After a revision of the paper, the decision is that the manuscript can be published after making the required corrections. It appears to have mistakes that should be corrected.

1.     There are some grammatical errors throughout the text. The authors should improve moderately the language of the manuscript.

2.     The author should unify the units in the text (for example Temperature °K)

3.     Tables should be renumbered appropriately (Example: Two tables contain a number of 3)

4.     The size of the writing titles of figures and tables should be unified.

5.     Table S1 is not reported in the text.

6.     On what basis was the concentration C0 = 0.125 mg/ml chosen to study the effect of pH and thermodynamic parameters on the absorption performance in phosphate-citrate buffer?

7.     The authors should added the used calibration curves with its coefficient of determination (R2).

8.     According to this study, in 2.8 Reusability studies: “For desorption, acetone (20 mL) was then added to the sorbent, and the mixture was stirred for 4 h.”  It is known that acetone evaporates easily. So how will the author justify that the volume does not change knowing that the Blending process lasts 4 hours?

9.     The author should verify the magnification units in Figure 3 (Figure 3. SEM (a, b) and EDX (c,d) of sample 3d).

Comments on the Quality of English Language

Some editing (moderate editing) for the English language is required 

Round 2

Reviewer 3 Report

Comments and Suggestions for Authors

Authors addressed my comments as requested